# The Paradoxical Effect of Living Alone on Cognitive Reserve and Mild Cognitive Impairment among Women Aged 60+ in Mexico City

**DOI:** 10.3390/ijerph182010939

**Published:** 2021-10-18

**Authors:** Antonio R. Villa, Elsa Guerrero, Ana M. Villa, Rosalinda Sánchez-Arenas, María Araceli Ortiz-Rodríguez, Vania Contreras-Sánchez, María Alonso-Catalán, Benjamín Guerrero-López, Ingrid Vargas-Huicochea, Germán E. Fajardo-Dolci, Claudia Díaz-Olavarrieta

**Affiliations:** 1Research Division, Faculty of Medicine, National Autonomous University of Mexico, 3000 Ave. Universidad, Copilco Universidad, Coyoacán, Mexico City 04510, Mexico; avilla@unam.mx (A.R.V.); nut.avilla@gmail.com (A.M.V.); 2University Program of Health Research, National Autonomous University of Mexico, 3000 Ave. Universidad, Copilco Universidad, Coyoacán, Mexico City 04510, Mexico; elsaguerrero2013@gmail.com; 3Epidemiology and Health Services Research Unit, CMN Siglo XXI, Mexican Institute of Social Security, Mexico City 06720, Mexico; felicitasarenas@gmail.com; 4Faculty of Nutrition, Autonomous University of the State of Morelos, Cuernavaca 62350, Mexico; araceli.ortiz@uaem.mx; 5Department of Psychiatry and Mental Health, Faculty of Medicine, National Autonomous University of Mexico, 3000 Ave. Universidad, Copilco Universidad, Coyoacán, Mexico City 04510, Mexico; vaniacontreras@comunidad.unam.mx (V.C.-S.); mariaalonsocatalan@gmail.com (M.A.-C.); jguerrero@facmed.unam.mx (B.G.-L.); vargashuicochea.ingrid@gmail.com (I.V.-H.); 6Faculty of Medicine, National Autonomous University of Mexico, 3000 Ave. Universidad, Copilco Universidad, Coyoacán, Mexico City 04510, Mexico; german.fajardo@unam.mx

**Keywords:** cognitive reserve, living alone, cognitive impairment, elderly women, Mexico

## Abstract

An elderly person who lives alone must often be autonomous and self-sufficient in daily living activities. We explored if living alone and marital status were associated with mild cognitive impairment and low cognitive reserve in a sample of Mexican women aged 60+ attending continuing education courses using a cross-sectional design. Objective cognitive functions were assessed using the MMSE and Blessed Dementia Scale. We administered the Cognitive Reserve Questionnaire. Independence skills were assessed with the Katz index and Lawton index. Multivariate logistic regression analysis was used. We recruited 269 participants ($\bar{x}$ = 69.0 ± 5.8 years). Single, widowed, separated, and divorced women comprised 73% of the participants. A third lived alone and 84% had completed high school. Mild cognitive deficit was observed among 24.5–29.0%; the upper range for cognitive reserve was 61.7%. Living alone versus living with someone was associated with cognitive impairment (OR = 0.51, *p* = 0.04) and with low to medium cognitive reserve (OR = 0.51, *p* = 0.02) after adjusting for confounding variables. Living alone was an independent factor associated with a lower probability of displaying mild cognitive impairment and a higher probability of displaying high cognitive reserve. Women living alone in this study had a more robust cognitive framework and had built their own support networks.

## 1. Introduction

In 2000, Mexico’s annual growth rate among the elderly was 5.1%, a figure that if kept stable would place the current older adult population at 7.6%. If this were to double every 19 years, by 2050, this age group will represent 21% of the total population [1]. In addition, for every 100 young adults (under 15 years), Mexico City will have approximately 209.7 inhabitants aged 65 and over [2].

Among the elderly, a supportive social network has a positive effect on their emotional wellbeing and self-esteem [3,4]. Lack of social support has also been documented as a consistent risk factor for poor health and quality of life among those who live alone [5]. However, a person who lives alone may also need to be autonomous and self-sufficient in daily living activities, both basic and instrumented [6,7]; the latter requires the need to develop more robust memory skills, which in turn brings about a higher cognitive reserve [8,9]. Cognitive reserve (CR) is defined as the ability a person has to adapt to age-related brain changes or damage caused by certain brain pathologies [10]. It is a difficult to measure construct; education and verbal intelligence have been widely used as proxies of CR, since direct measurement of an individual’ s CR is elusive [11]. CR is construed as an active skill because it is dynamic and can be modified by circumstances and cognitive experiences. It is determined by several modifiable factors, including occupational attainment, engagement in leisure activities, physical activity, social engagement, or brain-challenging tasks. CR has been shown to mitigate the negative impact of aging on cognitive function and it is worth noting that there are modifiable factors that can increase it [12].

Gender, a history of chronic disease, displaying cognitive and functional impairment, personality traits, experiencing stressful life events, family history of depression, experiencing sensory losses, social losses, insufficient physical activity, increased levels of dependency, lack of support, and loneliness are among the predisposing factors associated with depression [13]. Caring for an ailing partner has also been shown to have an impact on the burden of care, as this increases over time. This burden is greater for female than for male caregivers. However, psychosocial stressors also increase the severity of caregiving activities among both genders. [14,15].

Some individuals are more vulnerable to developing feelings of loneliness in response to environmental triggers than others [16]. Elderly men face widowhood under more challenging circumstances compared to elderly women, especially in traditional contexts such as Mexico’s, where the burden of household chores falls mainly on women and where a generational effect is also markedly prevalent [17,18,19]. A widowed woman may have the ability and resources to acquire the adaptative tools that living alone entails [20,21]. It is therefore important to make a distinction between experiencing loneliness and living alone. The first represents a feeling of abandonment and lack of contact with someone, while the second implies autonomy. Loneliness has been associated with an increased predisposition to illness, greater cognitive decline, and even premature mortality. On the other hand, people living alone can seek and receive emotional support outside the home and build their own support networks [22,23].

We explored if living alone in a megacity such as Mexico City (population: 21,782,000) or living with someone was associated with displaying mild cognitive impairment and low cognitive reserve. We surveyed a sample of Mexican women aged 60 and older who had on average completed high school or college.

## 2. Materials and Methods

We carried out a cross-sectional study among elderly women who were enrolled in the University of the Elderly (*Universidad de la Tercera Edad*), a public educational institution in Mexico City that offers continuing education courses. During 2017–2019, we recruited subjects in a consecutive way as they agreed to participate in the study and signed a written informed consent form. All interviews were conducted face to face. The approximate student enrollment at the time the study was fielded was estimated at 2000. The main reason women declined to participate was that the study instruments took on average 1.5 h to administer and they did not have sufficient time to respond.

The inclusion criteria were being 60 years or older, being female, not being totally deaf or blind, and not presenting with severe mental illness (i.e., schizophrenia) or cognitive decline (i.e., dementia). Participants needed to be able to respond to the survey questions by themselves without the help of a carer (caregiver) or proxy. Those who did not comply, did not complete the survey, or did not agree to participate were excluded.

The sociodemographic variables in the survey included age, marital status (single, married, cohabitating, widowed, separated, or divorced), the person or people that the participant was living with (alone, with a partner, with offspring, with another relative or a non-relative), current occupation (unemployed, homemaker, worker, retired, or receiving public funds (*pensionado*), and other), the financial independence of the participant (does or does not get any financial support), comorbidities (overweight, obesity, diabetes, systemic arterial hypertension, vascular cerebral disease, chronic pulmonary disease, hypothyroidism, history of cancer, current cancer, chronic renal disease, or depression).

Memory Cognitive Reserve was measured using the Cognitive Reserve Questionnaire with the following dimensions [24,25]: (i) academic background (primary, junior, and senior high-school, college/graduate school), (ii) parents’ academic background (can read and write, junior and high-school or college/graduate school), (iii) continuing education courses (did not attend, attended 1 to 2, 3 to 5, and >5), (iv) prior occupation (administrative, management, and executive level), (v) musical training (does not play at all, plays a little, received formal musical training), (vi) languages (speaks 1 or more aside from mother tongue, 2, 3, >3),(vii) reading habit measured by number of books read per year (never, occasionally, 2–5 a year, 6–10 a year, >10 a year), and (viii) plays memory-challenge games (never, occasionally, regularly). The final cognitive reserve classification comprised four categories: low range (≤6 points), low-medium range (7–9 points), high-medium range (10–14 points), and upper range (≥15 points). In this study, we added the lowest three categories that were considered as having low cognitive reserve: the lower range in addition to the medium-low range plus the medium-upper range (≤14 points). These values were compared with the upper range category (≥15 points). We decided this because the figures in the first categories were few and we had to create a dichotomous variable.

Objective cognitive functions were assessed using (i) Folstein´s Mini Mental State Evaluation (MMSE), with the following cut-off scores: normal ≥27, suspected cognitive impairment 25–26, and mild cognitive impairment ≤24 [26,27]; (ii) the Blessed Dementia Scale, which measures changes in daily living skills and personal habits (in personality and behavior it is also used to screen for dementia, with the following cut-off scores: normal (<4), mild impairment (4 to 9), moderate (from 10 to 14), and severe (>15) [28]); and (iii) the Clock Test, which assesses visuospatial abilities, visual motor memory programming, and other abilities with the following cut-off scores: unaltered (0 mistakes), slightly altered (1–2 mistakes), moderate (3–4 mistakes), and severe (5 or more mistakes). This test has been shown to correlate with scores in the MMSE [29].

Basic activities of daily living (BADL) were measured using the Katz index as preserved (score = 6 shows total self-feeding, mobility, continence, toileting, dressing, and bathing abilities) or altered in one or more functions [30]. We administered the Lawton instrumental activities of daily living (IADL), which assesses the ability to use the phone, go shopping, prepare meals, perform household chores, use public transportation, and handle prescription drugs and money, and is classified as total independence (=8), slight (6–7), moderate (4–5), mild (2–3), and total dependency (0–1) [31]. In addition to this, gait speed was calculated by measuring the time in seconds it took the patient to walk 8 m in a straight line (4 m back and forth) [32,33]^.^

The statistical analysis included the description of continuous variables through the mean and standard deviation as well as categorical variables through relative frequencies. The comparison of independent variables (sociodemographic, comorbidities, cognitive functions, functionality, and gait speed) among women with low cognitive reserve compared with participants that scored in the normal range in cognitive reserve was performed through logistic regression analysis deriving odds ratios (OR) by the exponential of regression coefficients. We reported the values of statistical significance with 95% confidence intervals and *p*-values. Multivariate models were built to test the main effect of living alone versus living with someone on cognitive impairment and cognitive reserve after adjusting by confounding variables. All analyses were carried out with SPSS/PC v25.0. The research protocol was submitted and registered by the Institutional Review Board (Research and Ethics Committee) at the Faculty of Medicine of the National Autonomous University of Mexico (FMED/CI/GRD/014/2011).

## 3. Results

We collected information on 269 women aged 60 and older. The mean age was 69.0 ± 5.8 years (minimum 60 to maximum 86). From the total sample, women that were single, widowed, separated, and divorced comprised 73%. A third of the participants (33%) lived alone, and this was our main effect variable to test. A total of 225 women (84%) had a high level of schooling (had completed high school, college, or a graduate degree). They were currently retired (58%) or homemakers (32%) and the majority received non-governmental financial aid (87%). Their prior work history included administrative (33%), management (32%), and executive level jobs (27%). Among the prevalence of chronic diseases, overweight and obesity were the most common (both added up to 71%), followed by hypertension (36%), hypothyroidism (17%), and diabetes (13%). This latter figure is lower than the one expected for their age group in Mexico. The prevalence of mild or suspected cognitive impairment on the MMSE was 29%. The prevalence of cognitive impairment (low, moderate, and severe) with the Blessed Orientation Memory Concentration Test was 24.5%. According to the Cognitive Reserve Questionnaire, 62% of women scored in the highest category (upper range) of cognitive reserve. Two-thirds reported reading as their main hobby. Over 40% of the participants frequently solved memory-challenge games, more than half played a musical instrument, and a third spoke two or more languages. It is worth noting the percentage of women that fulfilled Fried’s criteria for prefrailty (76%) and frailty (13%) despite having a reasonably sound health status. The percentage of women with total independence for instrumental activities of daily living was 89%, while 36% reported some degree of dependence when carrying out basic activities of daily living (see Table 1). 

Living alone had a statistical trend of association with a lower probability of displaying mild, moderate, or severe impairment as measured by the Blessed Dementia Scale (OR = 0.52, *p* = 0.05) as well as with a lower probability of having cognitive impairment or suspected cognitive impairment as measured by the MMSE (OR = 0.57, *p* = 0.07). The above results seem to indicate that a woman in our study who lived alone had 92% (inverse value of 0.52) and 75% (inverse value of 0.57), respectively, lower probability of having some degree of cognitive impairment compared with a woman who lived with someone. Furthermore, living alone was also statistically associated with a lower probability of having a low, medium-low, or medium-high range of cognitive reserve (OR = 0.52, *p* = 0.02), namely, a 92% lower probability of displaying decreased cognitive reserve. The clock test did not yield statistical significance (see Table 2).

When we compared the variables associated with social support, mild or moderate physical activities including walking outside the home, perception of satisfaction and memory status compared to their peers, and the prevalence of comorbidity, we did not find statistically significant differences among women who lived alone compared to those who lived with someone. There was a trend towards being more physically active (walking outside the home or gardening) among women who lived with someone and less physically active among women who lived alone. However, there was a significant difference in the percentage of women who lived alone and were primary caregivers (8%) compared to women who lived with someone and cared for someone (21%) (*p* = 0.008) (see Table 3).

For further analysis, we tested the main effect of low cognitive reserve associated with the probability of displaying cognitive impairment with an OR = 1.9, 95% CI: 1.05–3.43, and *p* = 0.03, regardless of the effect of age, living alone, functional dependence (measured by the Lawton scale), schooling (had completed college or more vs. an incomplete college degree or less), and caring for someone.

Table 4 shows that living alone is an independent factor associated with a lower probability of displaying mild, moderate, and severe impairment, and low, medium-low, and medium-high cognitive reserve. The main effect of living alone was adjusted in multivariate models by age (years), the number of Fried’s criteria for frailty syndrome, history of stroke, fatigue (by self-report), and gait speed (seconds). 

## 4. Discussion

Our results show that in our sample of Mexican elderly women, living alone is an independent factor associated with a lower probability of displaying mild cognitive impairment as assessed by two standardized international tests and with a higher probability of showing high cognitive reserve. When we tested the main effect of low cognitive reserve on the probability of displaying cognitive impairment, we found that it was a risk factor independently associated with different covariates, including living alone. Women that were single, widowed, separated, or divorced did not show an association with the same variables of cognitive decline or cognitive reserve. In our study, marital status did not necessarily imply women lived alone; for example, they could report being widowed and still live with a family member. However, women that reported being widowed, divorced, and separated did not have a higher risk of displaying mild cognitive impairment compared with married women. We may speculate that a woman who lives alone comprises one factor representing different dimensions [20]. They are able and used to carrying out household chores by themselves, and they may have overcome widowhood and/or divorce with perhaps more resilience than men [19,20,34,35]. These conditions may favor their sense of autonomy, independence, and functionality, and thus increase their cognitive reserve and decrease their risk of developing cognitive impairment [8,36]. 

Different social theories have been put forth to explain the ageing process [37]. Successful ageing and healthy ageing are two well-known examples. However, in 2002 the World Health Organization (WHO) made a call to consider another construct, that of active ageing [38]. Under this paradigm, active ageing entails the optimization of resources and opportunities to further a person’s health which in turn will translate into an enhanced quality of life as we age. Because ours is an aging world, it is of paramount importance that we find the mechanisms that allow people not to stagnate. Societies and health care systems need to recognize what their aging population’s needs are and satisfy them based on the complexities involved in growing old. Furthermore, this process could potentiate different abilities, including physical, social, and mental, both at the individual and collective levels. Our results seem to highlight the ways in which social participation becomes relevant for women and men to build supportive social networks by attending continuing education courses with a curriculum that also included arts and drama. As Teater and Chonody illustrated [37], active ageing is multidimensional and includes “being active and participating in social, economic, cultural, spiritual, and civic issues”. Therefore, our findings seem to indicate that women in our sample adhere to WHO’s definition of active, successful, and healthy ageing. There is evidence to suggest that social isolation is also associated with higher mortality (more prevalent among widowed men than widowed women) and that the effect of long-term isolation increases the risk of depression and dementia. As an older person ages, their social networks decrease, reducing their cognitive stimulus and lowering their cognitive reserve. Thus, the lack of social interaction together with loneliness have negative effects on cognitive reserve and impairment. This interaction could be mediated by the capacity for neuronal plasticity in analogous pathways at the level of the hippocampus and the prefrontal region [39]. 

Our study has several limitations: ours is a sample with characteristics that differ from the “average” Mexican woman aged 60 and older. According to the 2020 National Population and Housing Census, there was a total of 8,276,286 women aged 60+. Of them, only 11% (892,107) had completed higher education (college or graduate school). In our study, participants that comprised this level of educational attainment was 47.9% [40]. It was also a sample of reasonably healthy elderly women whose prevalence of chronic diseases was lower than the that reported at the national level for similarly aged women. According to the 2018–19 National Survey of Nutrition and Health of Mexico [41]: the prevalence of diabetes was 27.1% versus 12.7% in our study, hypertension was 47.8% versus 36.4%, overweight was 38.7% versus 50.8%, and obesity was 33.0% versus 19.9%. We also observed a difference in the functional dependence in instrumental activities of daily living of our participants compared to the one reported in the 2012 National Survey of Nutrition and Health of Mexico [42]: 10.8% versus 28.4%. However, our study sample had a higher prevalence in dependence of basic activities of daily living than reported in the latter survey: 36.4% versus 29.6%. It is also important to note that our subject’s mean age was 69 years. These factors may have influenced their memory skills and high cognitive reserve. Nevertheless, we were able to find statistically significant differences between women living alone versus women who reported living with someone as an associated independent factor. It is also possible that women with more independence may have had prior professional development that contributed to this, and while in our study only seven participants reported having an executive level job, most held professional positions (administrative or middle management); thus, we assume the probability of bias is low. However, we are aware that our results cannot be generalized to older women in Mexico City since ours is a selected sample with the characteristics described above. Our choice of screening instruments—the MMSE, the Blessed Dementia Scale, and the Clock Test, in terms of its sensitivity to detect cognitive impairment in a population without severe memory pathology—have been described in the literature and could be considered another study limitation. Our study design (cross-sectional) did not allow us to accurately distinguish whether women who lived with someone displayed cognitive decline or were physically frail and thus required assistance from a caregiver, unlike women who lived alone who were healthier and more independent. However, we think this possibility is not supported by our results, which showed that women who lived alone were not different from those who lived with someone in terms of sociodemographic characteristics, social support, perception of health, and comorbidities. In daily routines such as physical activities (walking outside the home, gardening, etc.) women who lived alone showed a trend towards statistical significance in performing better than women who lived with someone. Being a primary caregiver also stood out when we compared both groups of women; it was higher (21%) among women who lived with someone compared to those who lived alone (8%). This finding suggests that women who live with someone and are primary caregivers must be physically and cognitively fit, so this may not be a group displaying a bias towards a greater deficit in overall functioning.

## 5. Conclusions

While our sample of elderly women had higher levels of schooling compared to Mexico’s national average of 9.2 years, [43] had a low prevalence of some of the most common and epidemic chronic conditions that plague this country (diabetes, overweight, and obesity) [41,42], and belonged to a middle income bracket, living alone was an independent factor associated with a lower probability of displaying mild cognitive impairment assessed with a higher probability of showing high cognitive reserve. Our study participants could be construed as a “privileged” sample in terms of social determinants of health; however, they were also self-motivated and disciplined to further their education by attending continuing education courses at a public university. They played a musical instrument, spoke more than one language, read often, played memory-challenge games, and most probably created a robust social network while attending these intellectual and artistic activities. When we enquired if they received financial aid from the government (in the form of a pension), a very small percentage did (13%), which meant they perhaps led frugal lives. Since we did not find statistical differences between women who lived alone and those who lived with someone in terms of the prevalence of chronic diseases and activity outside the home, except for having cared for someone in the last week, we can assume that the former group of women displayed a more robust cognitive framework and were able to build their own support networks.

## Figures and Tables

**Table 1 ijerph-18-10939-t001:** Characteristics of functionally independent study participants aged 60 years.

Variable	*n* = 269	
Age (years)	69.0 ± 5.8	
BMI (kg/m^2^)	27.1 ± 3.9	
Gait speed test for 8 m (seconds)	4.6 ± 1.1	
	**N°**	**%**
**Marital status**		
Single	109	40.5
Married or common-law	72	26.8
Widowed	64	23.8
Separated	11	4.1
Divorced	13	4.8
**Who does she currently live with?**		
Alone	88	33
Partner	71	26.6
Children	78	29.2
Other: relative	24	9
Other: non-relative	6	2.2
Did not respond	2	-
**Level of schooling**		
Elementary and middle school	44	16.4
High school	96	35.7
College	113	42
Graduate school	16	5.9
**Current occupation**		
Retired	156	58.2
Homemaker	86	32.1
Self employed	17	6.3
Unemployed	3	1.1
Other	7	2.2
**Financial dependency**		
Does not get aid	34	12.7
Gets aid	234	87.3
Did not respond	1	-
**Comorbidity**		
Overweight	135	50.8
Obesity	53	19.9
Diabetes	34	12.7
Hypertension	98	36.4
Cerebrovascular disease	9	3.3
Chronic obstructive pulmonary disease	11	4.1
Hypothyroidism	45	16.7
Cancer	28	10.5
Chronic kidney disease	5	1.9
Depression	15	5.6
**Exhaustion**		
Fatigue (by self-report)	37	13.8
**Cognitive Reserve Questionnaire (CRQ)**		
**Level of schooling**		
Elementary-middle school	44	16.4
High-School	96	35.7
College or Graduate school	129	47.9
**Parents´ level of schooling**		
Can read and write	19	7.1
Junior high-school	137	50.9
High-school or College	113	42
**Attended continuing education courses**		
None	51	19
1 to 2	7	2.6
3 to 5	23	8.6
>5	188	69.9
**Prior occupation (work history)**		
Administrative work	111	41.3
Middle management	85	31.6
Executive level	73	27.1
**Musical training**		
Does not play at all	118	43.9
Plays a little	140	52
Formal musical training	11	4.1
**Languages**		
1 (mother tongue only)	179	66.5
2	78	29
3	7	2.6
>3	5	1.9
**Reading activity**		
Never	33	12.3
Occasionally	56	20.8
2–5 books in one year	106	39.4
6–10 books in one year	45	16.7
>10 books in one year	29	10.8
**Memory-challenge games**		
Never plays	78	29
Occasionally	81	30.1
Regularly	110	40.9
**Cognitive Reserve Questionnaire (CRQ)**		
		15.2 ± 3.2
**Score**	1	0.4
	16	5.9
Low range	86	32
Medium-low range	166	61.7
Medium-high range		
Upper range		
**Cognitive Function (Mini Mental State Evaluation)**		
**Score**		
Normal		27.3 ± 2.1
Suspected cognitive impairment	191	71
Mild cognitive impairment	53	19.7
	25	9.3
**Blessed Dementia Scale**		
**Score**		2.02 ± 3.2
Normal	203	75.5
Mild impairment	56	20.8
Moderate impairment	8	3
Severe impairment	2	0.7
**Clock-Test**		
Unaltered	184	68.4
Mild alteration	50	18.6
Moderate alteration	30	11.2
Severe alteration	5	1.9
**Fried’s frailty criteria**		
Not frail	28	10.4
Prefrail	205	76.2
Frail	36	13.4
**Basic activities of daily living (Katz index)**		
Total independence	171	63.6
Some dependence	98	36.4
**Instrumental activities of daily living (Lawton index)**		
Total independence	240	89.2
Slight dependency	28	10.4
Moderate dependency	1	0.4
Moderate dependency		

**Table 2 ijerph-18-10939-t002:** Association between living alone, displaying cognitive impairment (MMSE and Blessed), and cognitive reserve.

Dependent Variable	Living Alone		Living with Someone		OR	95% CI	*p*-Value
	N = 88		N = 181				
	N°	%	N°	%			
**MMSE**							
Mild cognitive impairment or suspected cognitive impairment	19	21.6	59	32.6	0.57	0.31–1.03	0.07
**Blessed Dementia Scale**							
Mild, moderate, or severe impairment	15	17	51	28.2	0.52	0.28–0.99	0.05
**Cognitive Reserve Questionnaire**							
Low, medium-low, or medium-high range	25	28.4	78	43.1	0.52	0.30–0.91	0.02
**Clock test**							
Mild, moderate, or severe alteration	24	27.3	61	33.7	0.74	0.42–1.29	0.33

**Table 3 ijerph-18-10939-t003:** Association between living alone or with someone, social and physical activities, and comorbidity variables.

Variable	Lives Alone *n* = 88		Lives with Someone *n* = 181		*p*-Value
	N°	%	N°	%	
Do you have someone to take care you?	2	2.3	5	2.8	1.00
Do you receive financial aid?	79	89.8	155	86.1	0.44
Do you carry out activities outside the home?	85	97.7	177	98.3	0.66
Do you feel satisfied with life?	77	87.5	160	88.4	0.84
Do you prefer to stay home rather than go out?	14	15.9	34	18.8	0.61
Do you think you have more memory problems than most people?	9	10.2	14	7.7	0.49
Do you think other people have a better sense of well being compared to you?	3	3.4	7	3.9	1.00
In the last 7 days, did you carry out activities outside the home?	83	95.4	179	98.9	0.09
In the last 7 days, did you practice gardening at home?	19	21.8	60	33.1	0.06
In the last 7 days, were you someone else’s primary caregiver?	7	8.0	38	21.0	0.008
Do you suffer from hypertension?	29	33.0	69	38.1	0.42
Do you suffer from diabetes?	7	8.0	27	15.0	0.12
Do you suffer from chronic kidney disease?	3	3.4	2	1.1	0.34
Do you suffer from cerebrovascular disease?	3	3.4	6	3.3	1.00
Do you suffer from chronic obstructive pulmonary disease?	5	5.7	6	3.3	0.35
Do you suffer from hypothyroidism?	15	17.0	30	16.6	1.00

**Table 4 ijerph-18-10939-t004:** Multivariate models to test the main effect of living alone on cognitive impairment (MMSE and Blessed) and cognitive reserve.

Model 1—MMSE *			Model 2—Blessed *			Model 3—CRQ *		
Dependent variable: mild or suspected cognitive impairment vs. no cognitive impairment			Dependent variable: cognitive impairment (mild, moderate, and severe) vs. no cognitive impairment			Dependent variable: low, medium-low, and medium-high cognitive reserve vs. high cognitive reserve		
Main effect: living alone vs. living with someone			Main effect: living alone vs. living with someone			Main effect: living alone vs. living with someone		
OR	95% CI	*p*-value	OR	95% CI	*p*-value	OR	95% CI	*p*-value
0.57	0.31–1.03	0.06	0.51	0.26–0.97	0.04	0.51	0.29–0.89	0.02

* All models were adjusted for age (years) and number of Fried’s criteria for frailty syndrome, history of stroke, fatigue (by self-report), and gait speed (seconds).

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
