# Peer review of "The Paradoxical Effect of Living Alone on Cognitive Reserve and Mild Cognitive Impairment among Women Aged 60+ in Mexico City"

_ijerph, 2021, doi:10.3390/ijerph182010939_

Round 1

Reviewer 1 Report

Thank you to the authors for drawing the DAG, which I hope they found helpful.

2 comments:

  1. I believe that continuing education should be included in the DAG as an adjusted variable, as inclusion in the sample is conditional on this variable?
  2. It seems likely that living alone and cognitive function are predictors of attendance at continuing education. What is the reason for not including a causal pathway from living alone to continuing education, and from cognitive impairment to continuing education, in the DAG? 

Reviewer 2 Report

I have no comments to make. I would like to take this opportunity to congratulate the authors for their good work.

Author Response

Thank you for your comments.

Reviewer 3 Report

I am now satisfied with the way the authors addressed my previous comments.

Author Response

Thank you for your comments.

This manuscript is a resubmission of an earlier submission. The following is a list of the peer review reports and author responses from that submission.

Round 1

Reviewer 1 Report

Evaluation of the manuscript: The paradoxical effect of living alone on cognitive reserve and mild cognitive impairment among women aged 60+ in Mexico City

This is an interesting investigation highlighting the role of living alone on cognitive functioning in a sample of older women living in Mexico City. There are some merits in this study but also some limits. I will express my specific concerns below.

Major points

  • The fact that the sample was selected among people attending to “University of the Elderly” and that most of them had a high level of education biased the sample and made it non-representative of the whole population. The role of education (as one of the main proxies and components of cognitive reserve) was therefore difficult to be assessed. Although this limitation is in part covered in the discussion, there are some issues that originate from it. For instance, mostly due to this unbalance, the authors had to collapse together the lowest 75% of the cognitive reserve range (p. 3: “This decision was made because the figures in the first categories were few and we had to create a dichotomous variable”), although this made up a category with very heterogeneous individuals in terms of cognitive reserve level.
  • One of the main conclusions of this study, which seems to be corroborated at least statistically, was that “Living alone was an independent, protective factor, associated with a lower probability of displaying mild cognitive impairment and a higher probability of having high cognitive reserve” (see end of abstract). I am not sure about the interpretation of these results. In particular, I do not understand how living alone for an older adult might be a “protective factor” for having higher cognitive reserve. Cognitive reserve is indeed a construct that, given its definition, precedes (and does not follow) the possible events that later on led a person to live alone or together with others. In other words, the factors that contribute to cognitive reserve accumulate across the whole life-span and it is difficult to imagine that they may really depend on something that might arrive only late in life (e.g., living alone).
  • Possibly related to the previous point, the authors might also try to perform complementary analyses where cognitive reserve is treated as one of the predictive variables (and not only as the dependent one) together with the factor concerning “living alone” to disentangle their independent contribution to cognitive functioning. The cognitive reserve construct or similar ones have indeed been shown to successfully predict high level cognitive functioning (e.g., https://doi.org/10.3389/fpsyg.2018.00630; https://doi.org/10.3389/fnhum.2012.00327).

Minor points

  • It might be useful to discuss the present study in the light of some recent literature on loneliness and social distance e.g.,: https://doi.org/10.1016/j.tics.2020.05.016
  • Page 6 (from line 192): “Living alone was statistically associated with a lower probability of displaying mild, moderate, or severe impairment as measured by the Blessed Dementia Scale (OR=0.52, p=0.05) and a borderline association with a lower probability of having cognitive impairment or suspected cognitive impairment as measured by the MMSE (OR=0.57, p=0.07).” P values of 0.05 or higher are not statistically significant and should be more clearly indicated as tendencies.
  • The authors measured cognitive impairment mostly with MMSE, Blessed Dementia Scale and the clock test, which are not very sensitive measures of cognitive functioning in the non-pathological population. Although they were still able to find some effects regarding cognitive functioning, it would be useful to discuss this as a possible limit of the study.
  • 3. “...Clock-Test that assesses visuospatial abilities, care, understanding, planning, visual motor memory programming, inhibition and numeric knowledge (sequencing)”: that this test is able to specifically assess all these functions, given its coarse scoring, is hard to buy. I would tone down this sentence.
  • English should be checked for minor slips, e.g.:
    • Page 2, line 100: “written informed consent format”, change “format” with “form”.
    • Page 4, line 161: “All analysis was carried out with SPSS/PC v25.0” should read “All the analyses were carried out with SPSS/PC v25.0.”
    • Table 4: “All models were adjusted by”. Change “by” with “for”
    • Page 8, lines 260-261: “Therefore, we consider our findings seem to indicate”, delete “we consider”
    • Something is wrong with the sentence “women who lived alone showed a trend towards statistical significance in favor of women who lived with someone”: I would change “in favor of” with something like “in overcoming” of “in performing better than”.

Reviewer 2 Report

Conceptual error:
The crossover study does not allow causal inference, but only association between variables. Therefore, the authors should not claim from their results that living alone is a protective factor for cognitive decline.

Cognitive reserve is based on the education acquired by the individual (decades ago), throughout their lives, and does not change due to the current personal situation. 

Methodological flaws:
1. There is no data on the number of women who declined to participate. This could lead to a participation bias.
2. The Blessed Dementia Scale is a scoring tool based on a close informant interview, it was not designed as a self-reported assessment.

Reviewer 3 Report

Thank you for the opportunity to review this paper. The hypothesis is very interesting and worthy of investigation, and in general the study appears to have been well conducted and the methods are well described.

However, I am concerned that the study design may give rise to collider bias - this occurs when the exposure (living alone) and outcome (cognitive function/reserve) are both independent causes of a third variable (in this case attendance at continuing education courses), that the analysis is conditioned on (in this case through the sampling strategy). Collider bias can lead to biased estimates of association - see https://catalogofbias.org/biases/collider-bias/ https://www.ncbi.nlm.nih.gov/pmc/articles/PMC5837306/. 

The paper would be considerably strengthening if greater attention was paid to the risk of this type of bias - it may be helpful for example to generate a directed acyclic graph (DAG), e.g. see https://doi.org/10.1093/ije/dyaa213

I am not clear of the justification dichotomising the outcome variables - surely the continuous variables capture more variation?